# Physical networks from entropy-driven non-covalent interactions

Anthony C. Yu[1], Huada Lian[1,2], Xian Kong [2], Hector Lopez Hernandez[1], Jian Qin [2] & Eric A. Appel [1✉]

Physical networks typically employ enthalpy-dominated crosslinking interactions that become more dynamic at elevated temperatures, leading to network softening. Moreover, standard mathematical frameworks such as time-temperature superposition assume network softening and faster dynamics at elevated temperatures. Yet, deriving a mathematical framework connecting the crosslinking thermodynamics to the temperature-dependent viscoelasticity of physical networks suggests the possibility for entropy-driven crosslinking interactions to provide alternative temperature dependencies. This framework illustrates that temperature negligibly affects crosslink density in reported systems, but drastically influences crosslink dynamics. While the dissociation rate of enthalpy-driven crosslinks is accelerated at elevated temperatures, the dissociation rate of entropy-driven crosslinks is negligibly affected or even slowed under these conditions. Here we report an entropy-driven physical network based on polymer-nanoparticle interactions that exhibits mechanical properties that are invariant with temperature. These studies provide a foundation for designing and characterizing entropy-driven physical crosslinking motifs and demonstrate how these physical networks access thermal properties that are not observed in current physical networks.

[1] Department of Materials Science & Engineering, Stanford University, Stanford, CA, USA. [2] Department of Chemical Engineering, Stanford University, Stanford, CA, USA. ✉email: eappel@stanford.edu

Physically crosslinked networks instill many natural and synthetic materials with function-critical properties otherwise unobtainable with static covalent networks[1–14]. As an engineering motif, physical networks have provided continued utility for drug delivery[3–6,15–18], biomaterials[7,19–22], wearable electronics[11–13,23,24], and 3D printing[25–27] on account of their distinct dynamic and stimuli-responsive properties. While these materials are often required to exhibit precise mechanical properties over a range of operating temperatures such that thermal responsiveness must be considered in their design, they typically weaken (i.e., become more liquid-like, soften, and relax stress faster) with increasing temperature. Indeed, standard mathematical frameworks for describing temperature-dependent viscoelasticity of dynamically crosslinked materials such as time–temperature superposition assume network weakening at elevated temperatures[28–36].

An abundance of dynamic, non-covalent crosslinking interactions have been described for building physically crosslinked polymer networks, including ionic, host–guest, hydrogen bonding, peptide–peptide, and hydrophobic interactions[6,15]. Examination of the thermodynamics of individual non-covalent interactions typically employed for crosslinking in physical networks reveals them to be enthalpy-dominated binding reactions[28–30,33,37]. This inclination arises because engineered binding pairs have been historically designed to be highly specific, which are by nature enthalpy driven at the cost of entropy[38]. Indeed, host–guest crosslinks such as β-cyclodextrin/adamantane (CD/Ad) and cucurbit[8]uril/viologen/naphthalene (CB[8]/MV/ Np) are favorable due to a combination of steric fit, hydrogen bonding, electrostatic interactions, and a large enthalpic gain from the release of high energy water[29,37,39]. Hydrogen bonding interactions such as 2-ureido-4[1H]-pyrimidinone (Upy), which typically comprise arrays of multiple complementary hydrogen bonds, are also strongly enthalpy driven[40,41]. Furthermore, a study of ~100 protein–ligand complexes showed that most protein–ligand binding interactions are enthalpy-dominated[38]. The many physically crosslinked networks reported in the literature have therefore historically been designed exclusively with enthalpy-dominated crosslinking interactions.

In this work, we develop a mathematical relationship between crosslink interaction thermodynamics and bulk viscoelasticity of the resulting physical networks. We use this relationship to describe the thermally induced softening (i.e., decrease in shear storage modulus) observed in physical networks built by enthalpy-driven crosslinks and to propose the ability to form physical networks exhibiting mechanical properties with alternative temperature dependencies by employing entropy-driven crosslinks. We then report an entropy-driven physical network based on dynamic and multivalent polymer–nanoparticle (PNP) interactions to create physical hydrogels that exhibit temperature-independent viscoelasticity. Experiments and simulation show the entropy-driven thermodynamics of the PNP interactions, while numerous rheometry experiments elucidate the structural changes that lead to temperature-invariant mechanical properties. These studies provide a foundation for designing and characterizing entropy-driven physical crosslinking motifs and demonstrate how these physical networks access critical thermal properties that are not typically observed in current physical networks.

## Results

**Thermodynamics of non-covalent interactions.** Specific non-covalent binding interactions can be schematically generalized as reversible complementary binding reactions with an associated Gibb's free energy of binding ($\Delta G°$) and an equilibrium constant ($K_{eq}$) that is related to the ratio of the forward and reverse reaction rate constants ($k_a$ and $k_d$) (Fig. 1). The $\Delta G°$ comprises an

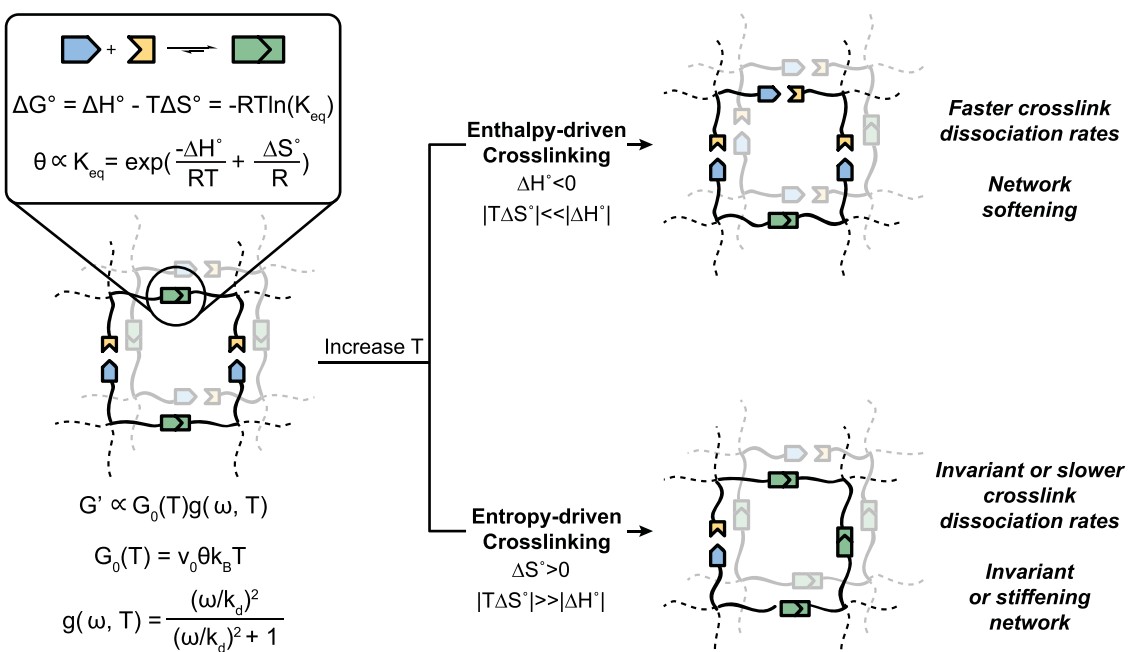

**Fig. 1 Enthalpy-driven and entropy-driven crosslinking in physical networks.** Schematic illustrating a network formed by physical interactions that can be individually symbolized as complementary binding interactions with corresponding thermodynamic constants. Two categories of binding interactions can be described—enthalpy-driven and entropy-driven interactions—which exhibit different responses to changes in temperature. The storage modulus ($G'$) of a physical network is represented by a plateau modulus ($G_0(T)$) and a frequency-dependent single-mode Maxwell term ($g(\omega,T)$). For simple physical networks, the relaxation constant is equivalent to the dissociation rate constant ($k_d$) of the complementary crosslinking interactions[55]. Enthalpy-driven crosslinks exhibit faster dissociation rates and network softening at elevated temperatures, while entropy-driven crosslinks dampen these temperature-induced changes in network relaxation, yielding either temperature-independent viscoelasticity or network stiffening.

enthalpy contribution ($\Delta H°$) and entropy contribution ($T\Delta S°$) and must be negative for a spontaneous binding interaction to occur under standard conditions. Two types of crosslinking can therefore be defined: (i) enthalpy driven whereby $\Delta H°$ is negative and the dominant contribution for binding, and (ii) entropy driven whereby $T\Delta S°$ is positive and the dominant contribution for binding (Fig. 1 and Supplementary Fig. 1).

Analysis of this complementary binding scheme suggests that the simplest crosslinking reactions that occur with defined interaction stoichiometries and specific binding architectures should be enthalpy driven. If the system is defined as two reactants interacting to form one product, then $\Delta S°_{system} < 0$ in the forward (binding) reaction, which means binding must be exothermic ($\Delta H° < 0$) for crosslinks to form spontaneously. Moreover, as temperature is increased, the influence of the $T\Delta S°$ term will increase, shifting the reaction toward the unbound states[38,42]. In physical networks, crosslinking interactions of this sort will become less favorable ($\Delta\Delta G° > 0$) with increasing temperatures, contributing to a commensurate decrease in network mechanical properties. Indeed, common physically crosslinked polymer networks based on calcium/alginate, CD/Ad, and CB[8]/MV/Np interactions exhibit enthalpy-driven thermodynamics ($\Delta H° < 0$; $T\Delta S° < 0$; $|\Delta H°| >> |T\Delta S°|$) and network softening at elevated temperatures[28–30,33,37] (Supplementary Table 1).

Despite the prevalence of enthalpy-driven behavior in physical networks, the complementary binding scheme also predicts the ability for entropy-driven crosslinks to form, where the $T\Delta S°$ term dominates the free energy change of the crosslinking interactions (Fig. 1 and Supplementary Fig. 1). While some entropy-driven interactions have been identified (e.g., desolvation events giving rise to large increases in translational entropy of solvent molecules or combinatorial entropy arising from distinct intra- and inter-particle binding of polymers), they are exceedingly rare in synthetic systems[43–45]. Adsorption of polymers onto surfaces, including the surfaces of particles, has been extensively explored[45–48] and some systems exhibit entropy-driven thermodynamics, theorized to be due to significant entropy gain as semiordered water molecules surrounding the components are dispelled upon binding[43,49]. Accordingly, we hypothesized that physical networks formed by entropy-driven PNP interactions, whereby high molecular weight polymers adsorb onto the surface of nanoparticles[43,46,49] (Fig. 2a), would exhibit temperature-invariant mechanical properties while retaining the valuable dynamic mechanical properties typical of transient networks.

## Development of entropy-driven non-covalent interactions.
Dynamic and multivalent PNP interactions[5,29,35] can be designed so that solvent molecules solvating both the polymer and the complementary NP surface are released into the bulk upon binding, consequently offering large gains in translational entropy (Fig. 2b–d). Moreover, the change in the translational entropy of a polymer chain upon binding to a surface is typically negligible when the polymer volume fraction is sufficiently high and the polymer length is larger than the interparticle distance[50,51]. If these conditions are met and the enthalpy of binding ($|\Delta H°|$) is small compared to the entropy contribution ($T\Delta S°$), then the crosslinking interaction is defined as entropy driven. Here we use dodecyl-modified hydroxypropylmethylcellulose (HPMC-$C_{12}$; 2 wt%) and polystyrene nanoparticles (PSNPs; $D_H \sim 57$ nm; 5 wt%) to form dynamic PNP networks whereby the complementary binding between the HPMC-$C_{12}$ and PSNPs immediately induces a cohesive network structure of overlapping PNP units (each PNP unit is comprised of a nanoparticle with an adsorbed polymer corona) with solid-like relaxation behavior upon mixing of the two

constituents (Fig. 2a)[52]. Isothermal titration calorimetry (ITC) experiments demonstrate that the adsorption of HPMC-$C_{12}$ onto PSNPs is strongly entropy driven ($\Delta S° > 0$; $|T\Delta S°| >> |\Delta H°|$) and are best represented by a 1:1 binding scheme due to the single inflection point of the titration curve (Fig. 2b, Supplementary Table 1, and Supplementary Fig. 2). The shape of the titration curve also indicates that the multiple interactions that occur as the polymer adsorbs onto the surface have similar affinities and $\Delta H°$ values, indicating that a single-site binding model is the most appropriate for calculating the binding thermodynamics for PNP interactions[53]. The positive entropy gain upon adsorption of the polymer to the nanoparticle surface was corroborated by simulation, where the free energy change for adsorption decreases as temperature increases from 290 K to 330 K (Fig. 2c and Supplementary Fig. 3). Interestingly, ITC shows that these PNP interactions are endothermic, meaning that the interaction between HPMC-$C_{12}$ and PSNPs is favorable solely due to the strong, positive entropy contribution (Fig. 2b).

## Thermodynamics of physical networks.
Transient network theory[54,55] provides a convenient way to relate crosslinking thermodynamics to bulk mechanical properties, where the shear storage modulus ($G'$) is represented as the product between a classical rubber elasticity term $G_0(T)$ and a frequency-dependent $g(\omega,T)$ term:

$$G'(\omega, T) \sim G_0(T)g(\omega, T) \tag{1}$$

$$g(\omega, T) = \frac{(\omega/k_d)^2}{(\omega/k_d)^2 + 1} \tag{2}$$

The plateau modulus, $G_0(T)$, is proportional to the fraction of bound crosslinks $\theta(T)$, the crosslink density $\nu_0$, and thermal energy $k_B T$ (Fig. 1 and Supplementary Fig. 1). The $g(\omega,T)$ term is a function of the frequency of applied stress and the rate constant for crosslink disengagement[55] (Figs. 1, 2a, Supplementary Fig. 1, and Supplementary Discussion). In the limit of one dominant relaxation mode, $g(\omega,T)$ reduces to a single-mode Maxwell model (Eq. 2), typically constructed using a general relaxation rate $\beta(T)$. Yet, Craig and coworkers[56] previously demonstrated that the disengagement rate constant of the physical crosslinks ($k_d$) was in good agreement with the Maxwell component disengagement rate and could be substituted to make superposition correlations. When interpreted as a bond lifetime ($1/k_d$), it is also apparent that $k_d$ is proportional to the terminal relaxation time of the networks where the rubbery plateau ends[14].

The fraction of bound crosslinks, $\theta(T)$, is directly proportional to $K_{eq}$ (Fig. 3b, Supplementary Fig. 1, and Equation 7). Notably, the temperature dependence of $K_{eq}$, and therefore $\theta$, lies on the enthalpy term, whereby the sign of $\Delta H°$ dictates the system's temperature dependence, while the sign and magnitude of $\Delta S°$ determines the system's temperature sensitivity (Fig. 3b, Supplementary Fig. 1, and Equation 6). These trends are illuminated when the reported thermodynamic parameters for various physical crosslinks are used to calculate $\theta(T)$ (Fig. 3c). While enthalpy-driven crosslinks, including CD/Ad and CB[8]/MV/Np, and entropy-driven PSNP/HPMC-$C_{12}$ interactions exhibit opposite trends in the temperature dependence of $\theta(T)$, the magnitude of these changes are negligible ($\Delta < 1\%$) across the entire accessible temperature range for all of the physical networks evaluated[28–30,33,37]. This observation suggests that temperature-dependent changes in network viscoelasticity observed in many systems are not caused by changes in effective crosslink density as offered by classical results[57]. In contrast, calculations of the temperature dependence of $k_d$ based on the thermodynamic parameters for these crosslinks indicate that $k_d$ changes orders of

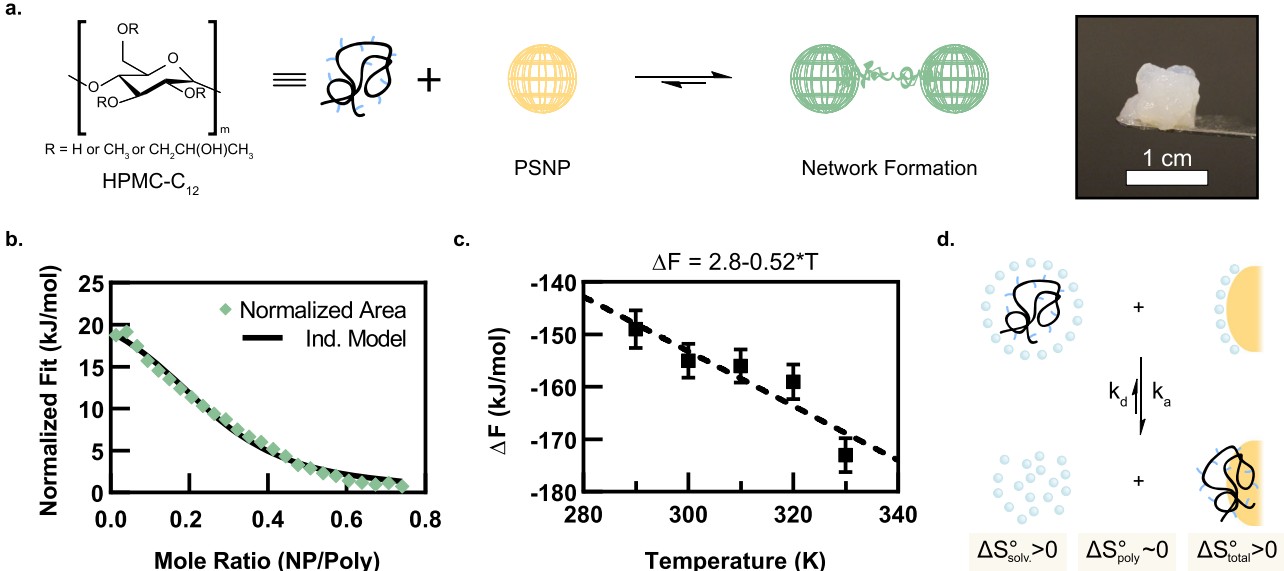

**Fig. 2 Polymer–nanoparticle (PNP) hydrogels exploiting entropy-driven interactions. a** Schematic of physical crosslinking between dodecyl-modified hydroxypropylmethylcellulose (HPMC-$C_{12}$) and polystyrene nanoparticles (PSNPs; $D_H = 57$ nm). The photograph shows the opaque hydrogel formed upon mixing of the polymer and nanoparticle constituents. **b** Normalized integrated heats of an incremental titration of PSNPs into a solution of HPMC-$C_{12}$ during an isothermal titration calorimetry (ITC) experiment. **c** Adsorption free energy changes calculated from a simulated HPMC-$C_{12}$ chain pulled off a complementary PSNP surface. Data shown as mean ± standard deviation. **d** Schematic illustrating the binding of a polymer chain onto a nanoparticle surface and the corresponding entropy changes of each component. PNP interactions form when the entropy change is large and positive, regardless of whether these interactions are endothermic or exothermic.

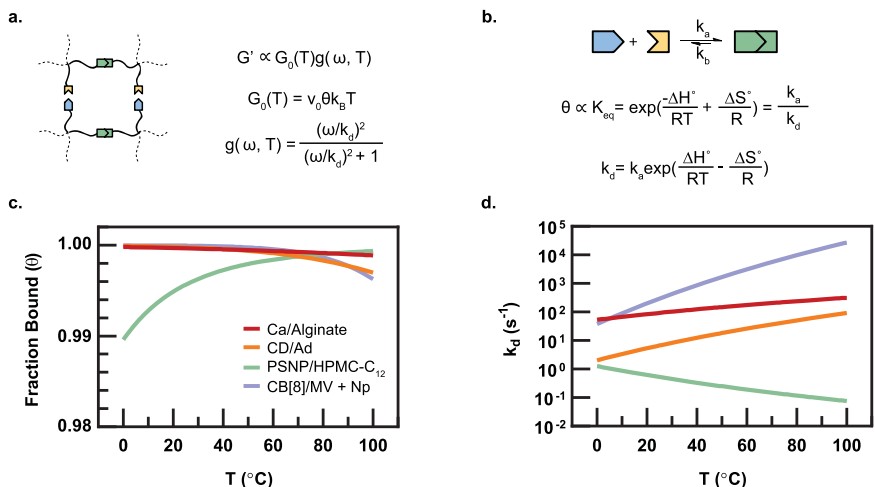

**Fig. 3 Temperature dependence of physical crosslinking interactions. a** Schematic of a physically crosslinked network and the associated equations used to describe the shear storage modulus. **b** Schematic of one binding interaction and the associated equations that relate the thermodynamics to the equilibrium constant ($K_{eq}$), the dissociation constant ($k_d$), and the fraction of bound crosslinks ($\theta$). **c** Calculated curves of $\theta$ versus temperature using the thermodynamic parameters for PSNP/HPMC-$C_{12}$ interactions and other notable physical interactions[28–30,33,36] using Equation 7 (Supplementary Discussion). **d** Calculated $k_d$ values for these same systems as a function of temperature from Equation 10 (Supplementary Discussion).

magnitude over the same temperature range (Fig. 3d). For enthalpy-driven crosslinks, $k_d$ increases with increasing temperature, indicating faster bond dissociation rates, while for entropy-driven PSNP/HPMC-$C_{12}$ interactions, $k_d$ decreases with increasing temperature, indicating slower bond dissociation rates. These observations suggest that temperature-induced changes in the mechanical properties of physical networks originate from perturbations in the crosslink bond lifetime dictated by $k_d$ and embodied in $g(\omega,T)$. Furthermore, the temperature dependence of $k_d$ suggests that entropy-driven crosslinks can imbue physical networks with temperature-invariant mechanical properties.

**Rheological characterization of PNP hydrogels**. Dynamic rheometry reveals that PSNP/HPMC-$C_{12}$ physical hydrogels show negligible changes in the magnitude and frequency dependence of the shear storage ($G'$) and loss ($G''$) moduli in the experimentally accessible temperature range (Fig. 4a and Supplementary Fig. 4). Amplitude sweeps demonstrate similar results, illustrating increased relative elasticity (i.e., a decrease in tan($\delta$), defined as $G''/G'$ as temperature is increased (Fig. 4b and Supplementary Fig. 5a). The yield stresses (defined here as the stress at which tan($\delta$) = 1 in the strain sweeps) are roughly ~800 Pa at all temperatures (Supplementary Fig. 5c). Hard sphere

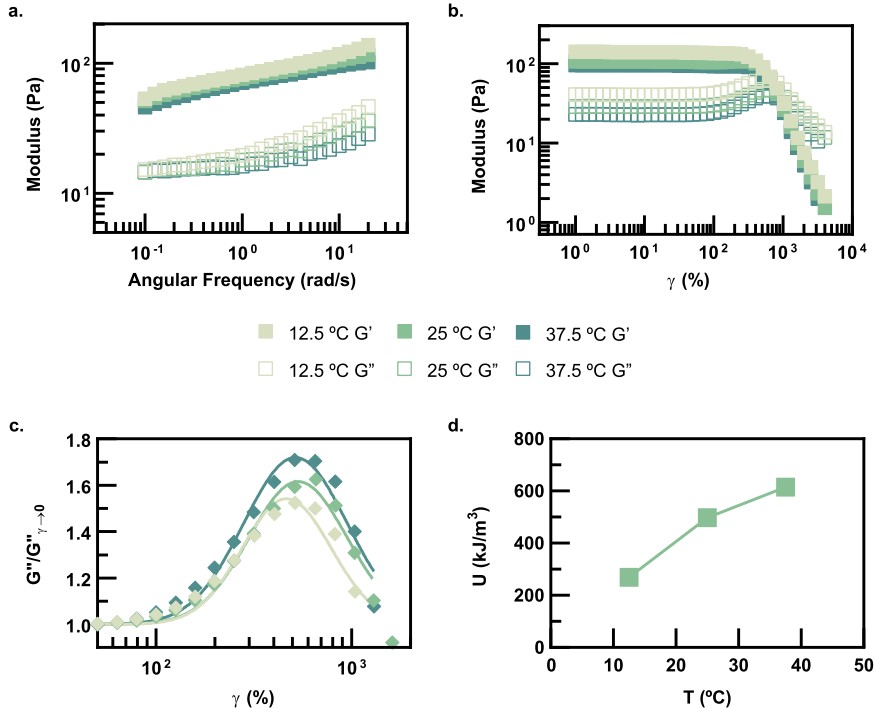

**Fig. 4 Dynamic rheometry of PSNP/HPMC-C12 hydrogels. a** Frequency-dependent linear rheometry ($\omega = 0.1$–20 rad/s) of PSNP/HPMC-$C_{12}$ PNP networks. **b** Strain-dependent rheometry ($\gamma = 1$–4000%) of the PNP gel at 12.5 °C, 25 °C, and 37.5 °C. **c** Normalized loss modulus peaks taken from the yielding point of the amplitude sweeps and fit to a log-normal distribution equation. **d** The area-under-the-curve of the normalized loss modulus peaks obtained from strain-dependent rheometry ($\gamma = 1$–4000%) of the PSNP/HPMC-$C_{12}$ PNP networks. These area-under-the-curve values are characteristic of the decaging energy of the physical crosslinking interactions and corroborate the predicted temperature-induced strengthening in PSNP/HPMC-$C_{12}$ PNP networks.

suspensions often exhibit yield stresses arising from entropic particle–particle interactions on the order of $k_BT/R^3$ ($R = 8.314$ J/mol/K), which is ~2 orders of magnitude less than the yield stresses measured here for the PSNP/HPMC-$C_{12}$ PNP networks[58] (Supplementary Fig. 5c). These observations suggest that the favorable yet dynamic PNP interactions (with binding thermodynamics determined from ITC) are critical to the elasticity and temperature responsiveness of the resulting hydrogel materials.

Further analysis of the amplitude sweeps provides insight into temperature-induced microstructural changes within the PSNP/HPMC-$C_{12}$ PNP networks (Fig. 4c, d). The characteristic $G''$ maximum at the crossover of $G'$ and $G''$ is indicative of local yielding ("uncaging") of PNP interactions[59–63]. In PNP hydrogels, uncaging occurs when PNP units locally yield through, for example, detachment of bridging polymer chains that are adsorbed onto the surfaces of adjacent nanoparticles, dissipating stress in the process. The area under the $G''$ peaks can be fit to a log-normal relation and the area under the curve represents the total energy dissipated per unit volume ($U$) for these local yielding events[59,64]. The normalized $G''$ plot illustrates that $U$ increases with temperature for these materials, suggesting that as temperature is elevated, the PNP structures require more energy to locally yield (Fig. 4d). This observation corroborates the entropy-driven thermodynamics of the HPMC-$C_{12}$ adsorption onto PSNPs, where it is expected that as temperature increases, the PNP interactions become stronger and less dynamic (Fig. 4d and Supplementary Fig. 5b). Overall, elevated temperatures make entropy-driven, mildly endothermic PSNP/HPMC-$C_{12}$ interactions slightly more favorable, which leads to physical networks with temperature-invariant moduli, more interconnected PNP structures, and more solid-like behavior.

Using calculated $k_d$ values, temperature-induced shifts in $G'/(v_0k_BT)$ can be calculated to neatly portray the effects of

enthalpy-driven and entropy-driven crosslinking observed in rheometry experiments (Fig. 5). Here, PSNP/HPMC-$C_{12}$ hydrogels were compared to physical networks formed by enthalpy-driven CB[8]/MV/Np interactions due to the use of similar cellulosic network polymers and the availability of thermodynamic and temperature-dependent rheological data[29–32]. For mildly endothermic, entropy-driven PNP networks, calculations suggest that elevated temperatures slow the dynamics of network relaxation and result in leftward shifts of the frequency-dependent rheology curve, signifying an extension of the terminal relaxation time (Fig. 5a). Given that the hydrogel formulations are in the rubbery plateau under the conditions evaluated, with nearly all crosslinks bound ($\theta \sim 1.00$), no substantial change is observed in $G'/(v_0k_BT)$. These trends are exemplified when the experimental data are overlaid onto the calculated curves for $G'/(v_0k_BT)$ values taken at a representative frequency of 10 rad/s (Fig. 5b). In contrast, the strongly enthalpy-driven CB[8]/MV/Np-based physical networks exhibit dramatic rightward shifts in the frequency-dependent rheology curve, representing a significant reduction in the terminal relaxation time, and a commensurate decrease in $G'/(v_0k_BT)$ at representative frequencies (Fig. 5c). These observations are predicted by the crosslinking thermodynamics and are in good agreement with experimental values of the frequency sweeps[30] (Fig. 5d).

To characterize the impact of temperature on the network relaxation dynamics, stress relaxation experiments were performed, and the data were fit to a single-mode Maxwell model to calculate network relaxation times ($\tau$) (Fig. 6). The model fits exhibit a negligible change in $\tau$ from 12.5 °C to 37.5 °C for the PSNP/HPMC-$C_{12}$ networks (Fig. 6a, b). In contrast, $\tau$ values estimated from the crossover frequency from frequency sweep plots reported in the literature for the enthalpy-driven CB[8]/MV/Np network[30] demonstrated a 92% decrease from

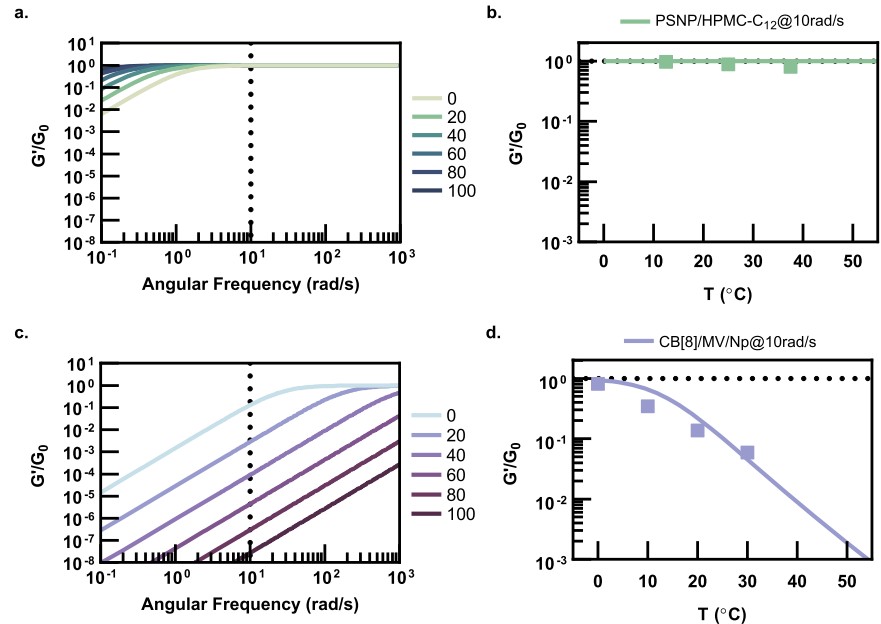

**Fig. 5 Effect of temperature-induced changes in $k_d$ on frequency-dependent rheometry. a** Calculated frequency-dependent $G'/G_0$ curves for PSNP/HPMC-$C_{12}$ gels. **b** Calculated $G'/G_0$ values taken at 10 rad/s show good agreement with experimental data and demonstrates negligible changes in modulus with elevated temperatures. **c** Calculated frequency-dependent $G'/G_0$ curves for representative enthalpy-driven CB[8]/MV/Np hydrogels using literature values for crosslinking thermodynamics[29–32]. **d** Calculated $G'/G_0$ values taken at 10 rad/s show good agreement with literature data and demonstrate a significant decrease in modulus as temperature is elevated.

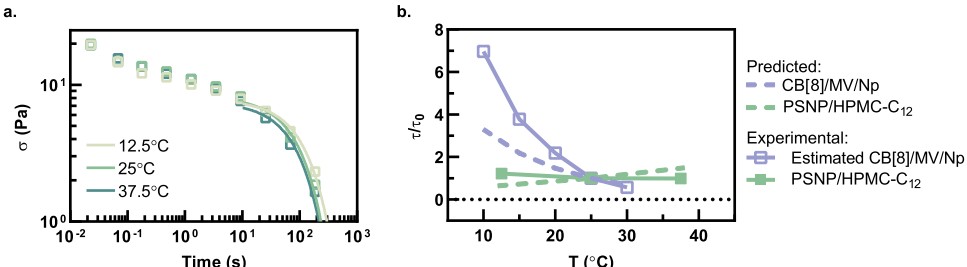

**Fig. 6 Temperature-dependent stress relaxation experiments. a** Stress relaxation experiments with single-mode Maxwell fits overlaid on the data points where only every 30th point is displayed for clarity. **b** Comparison of characteristic relaxation times between CB[8]/MV/Np and PSNP/HPMC-$C_{12}$ networks normalized to the relaxation time at 25 °C. Dotted lines represent calculated values from the model.

10 °C to 30 °C. The significant difference in temperature-dependent behavior between these two systems is further emphasized when $\tau$ is normalized to $\tau_0$ at 25 °C (Fig. 6b). Moreover, these experimental data match expectations based on the temperature dependence of the $k_d$ values for these two systems described above (Fig. 3d). Indeed, the $k_d$ values for the enthalpy-driven CB[8]/MV/Np system are expected to increase fivefold over the temperature range evaluated experimentally, commensurate with the 12-fold decrease in the normalized $\tau$ values observed experimentally (Fig. 6b). While the trends observed in the experimental data match theoretical expectations, the subtle deviation between the calculated and experimental normalized $\tau$ values for the CB[8]/MV/Np system likely arises from additional polymer-related relaxation events observed in this experimental system that are not captured in a single-mode Maxwell model[30]. In contrast, the $k_d$ values for the entropy-driven PSNP/HPMC-$C_{12}$ system are expected to decrease only twofold over the temperature range evaluated here, corresponding to the negligible change in the normalized $\tau$ values observed experimentally (Fig. 6b).

Experimental polymer hydrogel systems are expected to exhibit some additional relaxation mechanisms with temperature dependencies distinct from the non-covalent interactions responsible for crosslinking. Slow relaxing contributions such as trapped strands and entanglements arising in the concentration regime within the network are expected to exhibit decreased relaxation times at higher temperatures. While such behaviors will exacerbate the temperature-dependent relaxation of enthalpy-driven CB/MV/Np crosslinking interactions, the entropy-driven PSNP/HPMC-$C_{12}$ crosslinking interactions work to counteract these temperature-induced changes[65] (Supplementary Fig. 1). This hypothesis is corroborated by the negligible temperature dependence of stress relaxation in the PSNP/HPMC-$C_{12}$ networks ($E_a \sim 2.5k_BT$ at 25 °C) compared to the enthalpy-driven CB/MV/Np-based networks ($E_a \sim 36.5k_BT$ at 25 °C)[30]. Ultimately, these observations highlight the value of designing endothermic, entropy-driven crosslinking interactions to generate physical network materials exhibiting mechanics that are invariant within working temperature ranges.

## Discussion

Evaluation of the relationship between molecular scale thermodynamics and macroscopic temperature-dependent mechanical

behavior can provide insight into the design criteria enabling development of materials with application-specific temperature dependencies. Two terms are useful to model the mechanical behavior of these systems: (i) the plateau modulus $G_0(T)$ and (ii) the frequency dependence $g(\omega,T)$. Temperature-induced changes to the mechanics of physical network systems observed experimentally are often attributed to changes in the plateau modulus of the materials arising from changes in the crosslink density $\nu_0$ within the networks, which we represent as $\nu_0\theta(T)$ in our study. Yet, we show with several prominent experimental physical hydrogel systems that the proportion of bound crosslinks $\theta(T)$, which is dependent on $K_{eq}$ of the crosslinking interactions, does not change substantially over the entire accessible temperature range for hydrogels (i.e., 0–100 °C). Instead, the frequency dependence of the mechanics is implicated in significant temperature-induced changes in modulus values. We show that replacing the Arrhenius construction of the relaxation rate $\beta(T)$ with the enthalpic and entropic contributions to the dissociative rate constant, $k_d$, can capture temperature-induced changes in mechanical properties of distinct physically crosslinked networks.

Here we use this insight to develop an example of an entropy-driven physical hydrogel exhibiting mechanical properties that are nearly invariant to temperature within standard working temperature ranges. The agreement of the $G'/(\nu_0 k_B T)$ response to temperature between calculations and experimental values demonstrates that molecular scale crosslinking thermodynamics can inform macroscopic temperature-dependent behavior. Remarkably, a simplified picture of physical network formation comprising complementary binding interactions with one dominant relaxation mode (characteristic of a single-mode Maxwell model) matches the experimental behavior of enthalpy-driven and entropy-driven networks in three different rheological experiments (frequency sweeps, strain sweeps, and stress relaxation).

While the experimental system investigated here used endothermic, entropy-driven PNP interactions between PSNPs and HPMC-C$_{12}$ polymers, we speculate that any entropy-driven physical interactions can be leveraged to form networks exhibiting near temperature-invariant mechanical properties. Further, our mathematical modeling suggests that modulation of the degree to which entropy-driven interactions are exothermic or endothermic (Supplementary Fig. 1) provides a mechanism for fine-tuning the temperature-dependent mechanical properties of these networks. Collectively, abstracting physical networks as a collection of complementary binding interactions with corresponding thermodynamics provides a simple approach toward rationally incorporating a desired level of thermal responsiveness into engineered materials while maintaining valuable dynamic mechanical properties.

## Methods

**Materials**. Hydroxypropylmethylcellulose (HPMC; MW ~ 90 kDa), dodecyl isocyanate, *N,N*-Diisopropylethylamine (DIPEA), acetone, and anhydrous *N*-methyl-2-pyrrolidone (NMP) were obtained from Sigma-Aldrich. PSNPs (50 nm diameter) were obtained from Phosphorex Inc. Phosphate-buffered saline without calcium or magnesium (PBS) was obtained from Corning.

**HPMC-C$_{12}$ synthesis**. HPMC (1 g) was dissolved in NMP (40 mL) at 60 °C while stirring. Dodecyl isocyanate (187.5 μL) was dissolved in NMP (5 mL) and then added dropwise to the HPMC solution at room temperature while stirring. DIPEA (3–5 drops) was added and the reaction was left stirring at room temperature for ~16 h (overnight). The modified polymer was then precipitated from acetone, collected, re-dissolved in water, dialyzed in a Spectra/Por 3 dialysis membrane for 3–4 days against deionized water, and then isolated by lyophilization as a white solid. The HPMC-C$_{12}$ was then dissolved in PBS at 6 wt% to create a stock solution. The polymer was characterized by $^1$H-NMR (Inova 500 MHz) using DMSO-d6 as the solvent[49]. Final modification amounts were ~2 mol%, corresponding to 1 pendant dodecyl group for every 50 glucose units along the parent HPMC polymer chain.

**PSNP preparation**. PSNPs (Phosphorex; $R_H$ ~ 25 nm; 1 wt%) were concentrated by ultracentrifugation for 30–40 min at 3000 RCF using an Amicon Ultra-15 filter tube (10 kDa nominal molecular weight cut-off). After centrifugation, the PSNPs were diluted to a concentration of 75 mg/mL with PBS. Dynamic light scattering determined the PSNPs from this stock solution have a hydrodynamic diameter of $D_H$ ~ 57 nm (PD = 0.035).

**PNP hydrogel formation**. Gels were formed at 2 wt% HPMC-C$_{12}$ and 5 wt% PSNP by mixing the two components followed by centrifugation to remove bubbles. All samples were allowed to rest for ~16 h (overnight) before any rheometry.

**Isothermal titration calorimetry (ITC)**. All ITC experiments were performed using a Nano ITC Low Volume Calorimeter (TA Instruments) at 22 °C. Polymer solution (HPMC-C$_{12}$; MW ~ 22 kDa; 2.205 mg/mL) was placed in the sample cell and nanoparticle solution (PSNP; $D_H$ ~ 57 nm; 106.42 mg/mL) was injected using a 50 μL Hamilton syringe. The incremental titration was done over 26 injections (1.03 μL first injection; 1.95 μL all subsequent injections) at 720 s intervals. The subsequent heat data were fit using an independent binding model.

**Computational methods and simulation protocol**. The potentials of mean force (PMF; see below) for a HPMC-C$_{12}$ segment binding to a slab of polystyrene (PS) in water were investigated. The simulated HPMC-C$_{12}$ segments contained nine β-glucose units with two dodecyl chains connected at each end. The PS slab consisted of five flexible chains, each with a degree of polymerization of 50 and with the center of mass of the slab constrained in space. Both components were solvated by pure water using an SPC/E model in a 4.254 nm × 4.1753 nm × 35 nm simulation box. The system was modeled with cubic periodic boundary conditions using the GROMACS-53a6 force field with time steps of 2 fs. After energy minimization by the conjugate gradient method, the system was equilibrated in the NVT ensemble at the desired temperature. Temperature was maintained through velocity-rescaling with a time constant of 0.2 ps$^{-1}$. Electrostatic interactions were computed through particle-mesh Ewald summation. The list of non-bonded interactions was truncated at 12.0 Å.

**Potentials of mean force (PMF)**. We calculated PMF at five evenly spaced temperatures over 290 K to 330 K and chose the reaction coordinate as the $z$ distance between the center of mass of the PS slab and the center of mass of one dodecyl segment at an end of the HPMC-C$_{12}$ chain. The free energy vs. distance plot was graphed with every 30th point for clarity. The HPMC-C$_{12}$ chain and the PS slab spontaneously attract each other in water, decreasing the probability of spontaneous dissociation and making the sampling difficult. This challenge was overcome by dividing the reaction coordinate into a large number of windows and employing the umbrella sampling method, which introduces a biasing potential to improve sampling efficiency. The bias can subsequently be removed with the weighted histogram analysis method (WHAM). The initial configuration for umbrella sampling was formed by pulling one head of the HPMC-C$_{12}$ chain away from the PS slab along the $z$-axis at a rate of 0.001 nm/ps. A harmonic potential with a force constant $k = 1000$ kJ/mol/nm was applied to the reaction coordinate within each window. Then, 30 ns runs were performed for each sampling window. The last 15 ns was used in the WHAM analysis for all five cases. The error bar was estimated by the bootstrap method. Convergence of the umbrella sampling was shown in Supplementary Fig. 3. During simulations, the center of mass of the PS slab was held fixed with a strong harmonic constraint. This constraint only served to center the slab and had no impact on the results.

**Flow and dynamic rheometry**. All flow and dynamic rheometry measurements were performed on a torque-controlled DHR-2 Rheometer (TA Instruments) using a serrated 20 mm parallel plate geometry (Peltier plate steel) with a solvent well, a solvent trap, and a serrated plate bottom (Peltier plate steel). Samples were created at large enough volumes to split into three aliquots and each aliquot was used for one temperature (12.5 °C, 25 °C, 37.5 °C). In Supplementary Fig. 4b, c, samples were loaded and experiments were performed going from low to high temperatures and high to low temperatures. The temperature range was chosen because higher temperatures led to evaporation at the sample edges. Step shear experiments were performed by pre-shearing the material for 20 s at 0.1 s$^{-1}$ followed by 30 s at 1 s$^{-1}$. Data were then recorded for alternating periods of 120 s at 0.1 s$^{-1}$ and 30 s at 1 s$^{-1}$. Strain sweeps were performed from 1 to ~4000% strain at a frequency of 10 rad/s. Frequency sweeps were performed from 0.1 rad/s to 100 rad/s at a stress of 0.75 μN m. Frequency-dependent rheological values were plotted to 20 rad/s due to slip and inertial effects observed at higher frequencies (determined by eye and when phase values approached 180°). The torque value was chosen because it is well within the linear viscoelastic region determined from strain sweeps performed at each temperature (12.5 °C, 25 °C, 37.5 °C) at 10 rad/s and 0.5 rad/s. Temperature control was achieved using a Peltier stage. A log-normal equation was used to fit the $G''$ data gathered from the amplitude sweeps. The noise temperature ($X$) is calculated by: $X = 1 + (2/\pi)\delta$, where $\delta$ is the phase angle in the linear viscoelastic region.

**Stress relaxation rheometry.** All stress relaxation experiments were performed on a strain-controlled ARES-G2 (TA Instruments) using a 25 mm parallel plate geometry (Peltier plate steel) with a solvent trap. The sample was loaded at 12.5 °C and allowed to equilibrate until the axial force was constant before applying 5% strain and monitoring the stress over time. This strain value was chosen because it is well within the linear viscoelastic region of the material even at 37 °C. The gel was then brought to 25 °C and then 37.5 °C and equilibrated before performing the same experiment. The data were plotted (displaying every 30th point for clarity) as stress vs. time and a single-mode Maxwell model ($\sigma = \sigma_0 e^{\frac{-t}{\tau}}$) was used to fit points from $t = 10$–$420$ s. For Fig. 6b, $\tau_0$ is defined as the $\tau$ determined at 25 °C for each material.

## Data availability
The data that were produced and support the findings of this study are available from the corresponding author upon reasonable request.

## Code availability
The code used for the simulations that support the findings of this study are available from the corresponding author upon reasonable request.

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

## Acknowledgements

This work was financially supported by the Center for Human Systems Immunology with Bill & Melinda Gates Foundation (OPP1113682, OPP1211043) and the Stanford Bio-X Interdisciplinary Initiatives Seed Grants Program Round 9. A.C.Y. is grateful for the Kodak Fellowship (A.C.Y.). Some of this work was performed in the Stanford Nano Shared Facilities (SNSF), supported by the National Science Foundation under award ECCS-1542152. We thank Professor Sarah Heilshorn for use of her isothermal titration calorimeter. We thank Professor Andrew J. Spakowitz and Quinn MacPherson for discussion about the experiments and data. We also thank Dr. Snehashis Choudhury for the useful discussions.

## Author contributions

A.C.Y., H.L., X.K., J.Q. and E.A.A. designed the research; A.C.Y. performed the experimental work; H.L. and X.K. performed the simulation work. A.C.Y., H.L., X.K., H.L.H., J.Q. and E.A.A. analyzed the data. A.C.Y. and E.A.A. wrote the manuscript.

## Competing interests

The authors declare no competing interests.
