## [Peer Review File · Nature Communications]

REVIEWER COMMENTS

Reviewer #1 (Remarks to the Author):

The authors present a study on physically-crosslinked networks driven by entropic contributions. Overall this is an interesting study that deserves publication. A general criticism that I have is that I would have liked a stronger molecular characterization/component beyond the phenomenological description presented. Specificity is a desired feature in many of these systems since it allows for unique designs based on targeted applications. Therefore, a stronger characterization of the free energies based on the simulations (and the role of chemistry) would be welcome, at least in future studies by the authors.

Some specific comments:

- How is the entropy of HPMC-C12 affects the process? In Fig. 2D a seemingly random coil is always drawn but on hard surfaces the conformation tensor of long polymers is known to adjust so that it minimizes free energy (for example by increasing enthalpic interactions). Is this happening herein?
- Are the particles spherical and non-flexible? The slab in Suppl. Fig. 3 appears to have roughness.
- I believe past work from the authors have shown that HPMC-C12 has the strongest interactions with the PS particles relative to other similar chemistries; can this be rationalized within the concept of solvent entropic gain or do we need further contributions (from i.e. enthalpic interactions or linker entropic)?
- The authors mention in the supplementary information that 1:1 binding reaction is “a common and appropriate assumption for many systems”, can they support further this assumption for their system?
Specifically, would the simultaneous binding of two HPMC-C12 be described by the free energy profile shown? What is the role of concentration?

A technical comment:

- Accurate potential of mean force calculations are notoriously difficult (due to limitations in sampling) and the authors acknowledge it. However I suggest that the authors beyond the “pulling” experiment they perform a reverse process where they bring closer to the surface the HPMC-C12 chain (more sophisticated methods allow for replica exchanges). The profiles can be different (given potential challenges in calculating “reversible” work) but at least this will provide some error bars for the free energy shown in Fig.3 and support for the calculations.

Reviewer #2 (Remarks to the Author):

In the manuscript “Physical networks from entropy-driven non-covalent interactions” the authors clearly describe their goal of creating entropic rather than traditionally enthalpic driven physical networks in order to create hydrogels with a unique thermal response. Along the way, the authors lay out a mathematical relationship between cross-link interaction thermodynamics and bulk viscoelasticity. The authors combine simulations and experiments in an attempt to demonstrate the differences in viscoelastic response of their model materials to temperature. The manuscript is well written and easy to follow.

However, I do not recommend publishing this article in Nature Communications. Although the theory, and simulations are well reasoned, and I believe this research could eventually result in a novel contribution to the field of physical networks, the experimental results are much too limited and unconvincing. Below I outline my issues with the experimental results, and provide suggestions on how the authors could address these concerns.

The authors conduct a frequency sweep for a gel at 12.5, 25 and 37.5°C in Figure 4. The thesis of the paper demands that this experiment shows an increase in the network relaxation time with increasing temperature. This is not clearly observed at all from this data, and actually raises questions about the applicability of the developed model because the data is not characteristic of a single-mode Maxwell model, the foundation of their theoretical development. This plot is critical for the paper; however, it appears to rely on a single gel, raising questions of reproducibility. Additionally, in the protocol, it is unclear what the order of temperature testing was for this gel? Hot to cold or cold to hot? A critical control experiment is needed to show that this does not matter, and that their results are reproducible. Regardless, even if they can show that the results are reproducible, the results are currently extremely underwhelming and insufficient to support their overarching hypothesis. For example, the experimental temperature range is very narrow, just 25°C. The authors explain that this is their experimental limit, but do not provide a reason for this limitation. Is it evaporation and/or dehydration? If so, they authors could try to seal the edge of the rheometer tool with mineral oil to prevent evaporation. This should easily allow data to be obtained from at least 5°C to 35°C, but conceivably much higher to maybe 60-70°C. Hopefully the experimental results would be more convincing over this larger temperature range whereby an observable gel relaxation could be reached. Alternatively, why did the authors not try a series of step-strain tests at different temperatures, whereby the gel relaxation could more easily be observed and fitted to a single mode Maxwell model? As it stands, the presented rheological data does not look any different from other slow-relaxing polymer-particle hydrogels (entropically dominated or not).

Finally about Figure 4, it is unacceptable in its current form. As readers, we are supposed to be able to look at this data and easily track changes in the loss and storage moduli as a function of temperature; however, the resolution of the current plot is way too small, and the data symbols are way too big to discern any trends in these data. Again, my personal suggestion would be to try to measure and plot equivalent stress-relaxation data instead, since the proposed unique temperature-induced changes to gel relaxation time should be more easily confirmed.

Along the same line of reasoning; the lack of a single mode Maxwell model fit to the experimental frequency sweep data is a fatal flaw to this study. The idea that the material behaves according to this model is fundamental to the support of their theory, and currently appears to be unfounded. If the authors can in fact collect frequency sweep data (or stress-relaxation data) and successfully fit it to a single-mode Maxwell model, and actually show a clear increase in the network relaxation time as a function of increasing temperature, then the validation of the central hypothesis of the paper would be significantly enhanced.

[As a side-note; even if the authors can collect such data, I would be surprised if the data in fact fits a single-mode Maxwell model, given the multi-sticker nature of the PNP network crosslinks. Data from similar types of gels published in the literature at least suggests significant broadening of the

resulting relaxation spectra. This shouldn't change the overall support of the entropy-dominated crosslink behavior proposed by the authors though.]

Moving on to the other key experimental results in Figure 5, where the authors both compare their proposed entropic network with a known enthalpic network and relate these results to theory. A quick examination of their theoretical results show why this experiment has little value — the two networks have vastly different base relaxation times. In the entropic network, one observes little change in plateau moduli as a function of temperature, but that is predicted for both the entropic and enthalpic network if you remain in the plateau region far away from the relaxation time. In contrast, the enthalpic network is examined in the terminal relaxation region, so unsurprisingly there are large changes in the relative moduli as a function of temperature. In other words, if the dynamic shear measurements on the enthalpic network had instead been conducted at say 10^4 rad/s, then this materials viscoelastic moduli would also have been demonstrated to be temperature invariant. In essence, I do not see how this data looks any different from a comparison between “slow” and “fast” relaxing 'enthalpic' gel networks.

Instead, the theoretical predictions in Figure 5a actually clearly lay out what must be demonstrated experimentally to support the central hypothesis; an increase in relaxation time with increasing temperature. This behavior is predicted to be observed at low frequencies, which understandably can be hard to measure, but this frequency range needs to somehow be probed, either through a longer or higher temperature frequency sweep, or alternatively through another rheological measurement such as a stress relaxation, in order to clearly demonstrate the predicted unique entropy-dominated relationship between relaxation time and temperature.

Finally, a general note on the choice of vocabulary chosen throughout the text by the authors to describe the temperature-induced effects on the mechanical properties of the networks studied. Words such as ‘softening’ and ‘weakening’ needs to be treated with much more caution, as such words will otherwise become confusing terms given the thermodynamic framework within which this study operates. Without proper explanation, these terms become ambiguous and leave the reader hanging as to what the author’s actually mean, i.e. more liquid-like or networks with lower strength?

Reviewer #3 (Remarks to the Author):

I like this paper a lot. The idea that entropy can dominate in such situations is interesting but not without precedent. I will point you to the early work of Hooper and Schweizer (2005) where they showed that NP/polymer mixtures could phase separate under large enough attractive interaction due to bridging interactions. Why would bridging interactions give an entropic attraction ? A recent paper by Sciortino et al., ACS Nano (2020) goes to the heart of this mechanism, on admittedly a different system. So, please read these previous works and see how your results fit in with this picture. Otherwise a nice contribution to the current literature.

Sanat K Kumar

Reviewer #1 (Remarks to the Author):

The authors present a study on physically-crosslinked networks driven by entropic contributions. Overall this is an interesting study that deserves publication. A general criticism that I have is that I would have liked a stronger molecular characterization/component beyond the phenomenological description presented. Specificity is a desired feature in many of these systems since it allows for unique designs based on targeted applications. Therefore, a stronger characterization of the free energies based on the simulations (and the role of chemistry) would be welcome, at least in future studies by the authors.

We thank the reviewer for their helpful suggestions and comments. We have addressed each point below and have added additional discussion in the text.

Some specific comments:

- How is the entropy of HPMC-C12 affects the process? In Fig. 2D a seemingly random coil is always drawn but on hard surfaces the conformation tensor of long polymers is known to adjust so that it minimizes free energy (for example by increasing enthalpic interactions). Is this happening herein?

In the simulation, the HPMC-C₁₂ segment adjusts itself to lie on the PSNP surface. However, the simulation segment is certainly not representative of a long chain conformation that is likely to occur experimentally. Our “corona” picture suggests that the net entropy effect arises from both the entropy change of the large number of water molecules and the chain conformation. Since HPMC-C₁₂ chains are relatively stiff, with low configurational entropy to begin with, most of the net entropy effect likely comes from the water molecules. This dominance of entropy change due to the water molecules is similar to rod-like polymers such as chromosomes, where the main contribution of adsorption comes from the surrounding water molecules.

- Are the particles spherical and non-flexible? The slab in Suppl. Fig. 3 appears to have roughness.

The polystyrene slab used in the simulations consists of flexible polymer chains that are not constrained in space; however, the center of mass of the slab is constrained in space. This setup allows for a surface topography that most resembles the polymer nanoparticle. We have included these details in the methods section.

- I believe past work from the authors have shown that HPMC-C12 has the strongest interactions with the PS particles relative to other similar chemistries; can this be rationalized within the concept of solvent entropic gain or do we need further contributions (from i.e. enthalpic interactions or linker entropic)?

For robust (*e.g.*, $\tan(\delta) < 1$, low frequency-dependent moduli) gel formation we identified that beyond PNP interaction strength, the concentration of solids must be high enough for sufficient overlap of polymer coronas formed around the particles¹. In other words, in addition to the crosslinking interactions themselves, there is a geometrical factor that must be considered similar to any network system. For example, in a previous study¹, we observed that HPMC-C₆ polymer formulations did not form sufficiently large polymer coronas to span the interparticle spacing

between particles in gels formed at concentrations of 2 wt% polymer and 5 wt% PSNPs. The difference in corona height is most likely due to how the differently modified HPMC chains are solvated by the water, which we suspect is enthalpic by nature.

1. Yu, A. C., Smith, A. A. A. & Appel, E. A. Structural considerations for physical hydrogels based on polymer–nanoparticle interactions. *Molecular Systems Design & Engineering* **5**, 401-407, doi:10.1039/C9ME00120D (2020).

- The authors mention in the supplementary information that 1:1 binding reaction is “a common and appropriate assumption for many systems”, can they support further this assumption for their system? Specifically, would the simultaneous binding of two HPMC-C12 be described by the free energy profile shown? What is the role of concentration?

The reviewer brings up an excellent point about how we envision this picture specifically applies to HPMC-C₁₂ chains adsorbing onto the nanoparticle and whether the 1:1 binding model is appropriate. Representing crosslinking interactions in physically associative networks with an “extent of reacted groups” commonly leads to the same math as a 1:1 binding interaction. We use this same assumption due to its simplicity and prevalence in describing associative networks. We define the binding scheme as each monomer along an HPMC-C₁₂ chain adsorbing onto a site on the surface of the polystyrene nanoparticles and measure the thermodynamics of those interactions using ITC. In that sense, although there are multiple monomer components along an HPMC-C₁₂ chain interacting on the polystyrene surface, the overall thermodynamics is still best modelled by a single binding interaction as seen with the single inflection point in the ITC curve. While multiple interactions are likely occurring, the existence of this single inflection point indicates the interactions have similar binding affinities and ΔH values, which allows the curve to be fit to a 1:1, single-site binding model¹. If two HPMC-C₁₂ chains simultaneous bound to the polystyrene nanoparticle surface, we believe each interaction at the sub-chain scale, which is ultimately the interaction responsible for crosslinking, would still be described by the entropy-driven thermodynamics. At higher concentrations, we expect the surface of the nanoparticles to be saturated, eliminating the formation of additional PNP interactions and for the thermodynamics of additional interactions to be driven by polymer corona-polymer interactions. We have added additional discussion about the ITC results in the main text.

1. Le, V. H., Buscaglia, R., Chaires, J. B. & Lewis, E. A. Modeling complex equilibria in isothermal titration calorimetry experiments: Thermodynamic parameters estimation for a three-binding-site model. *Analytical Biochemistry* **434**, 233-241, doi:10.1016/j.ab.2012.11.030 (2013).

A technical comment:

- Accurate potential of mean force calculations are notoriously difficult (due to limitations in sampling) and the authors acknowledge it. However I suggest that the authors beyond the “pulling” experiment they perform a reverse process where they bring closer to the surface the HPMC-C12 chain (more sophisticated methods allow for replica exchanges). The profiles can be different (given potential challenges in calculating “reversible” work) but at least this will provide some error bars for the free energy shown in Fig.3 and support for the calculations.

We selected the binding state as the starting point for the free energy calculations because HPMC-C₁₂ and polystyrene surfaces naturally form strong attractive interactions. Further, the pulling experiment can save a considerable amount of time in the equilibrium stage of each window of umbrella sampling.

We also attempted to make the free energy calculation following the reverse process as recommended by the reviewer; however, the results showed that the reverse experiment cannot equilibrate to a fully adsorbed state with a trivial equilibrium MD simulation. The difficulties arise from the string-like structure and large configuration space of HPMC-C₁₂ in our system. Since we focus on the temperature dependence of free energy, we believe a detailed study on the choice of thermodynamic path is beyond the purpose of the current work. Instead, in order to demonstrate the convergence of PMF in the pulling experiment, we extended the umbrella sampling from 11 ns to 30 ns. For example, at 310K, the PMFs was constructed by using the last 15, 12, 10, and 7 ns converge consistently as shown by Supplementary Figure 3b.

Reviewer #2 (Remarks to the Author):

In the manuscript “Physical networks from entropy-driven non-covalent interactions” the authors clearly describe their goal of creating entropic rather than traditionally enthalpic driven physical networks in order to create hydrogels with a unique thermal response. Along the way, the authors lay out a mathematical relationship between cross-link interaction thermodynamics and bulk viscoelasticity. The authors combine simulations and experiments in an attempt to demonstrate the differences in viscoelastic response of their model materials to temperature. The manuscript is well written and easy to follow.

However, I do not recommend publishing this article in Nature Communications. Although the theory, and simulations are well reasoned, and I believe this research could eventually result in a novel contribution to the field of physical networks, the experimental results are much too limited and unconvincing. Below I outline my issues with the experimental results, and provide suggestions on how the authors could address these concerns.

We thank the reviewer for their kind and insightful responses to our work. We have addressed the issues brought up by the reviewer with additional experiments and discussion, which we outline in each point below.

The authors conduct a frequency sweep for a gel at 12.5, 25 and 37.5°C in Figure 4. The thesis of the paper demands that this experiment shows an increase in the network relaxation time with increasing temperature. This is not clearly observed at all from this data, and actually raises questions about the applicability of the developed model because the data is not characteristic of a single-mode Maxwell model, the foundation of their theoretical development. This plot is critical for the paper; however, it appears to rely on a single gel, raising questions of reproducibility.

We have included the control experiments in Supplementary Fig. 5 to illustrate the temperature behavior is reproducible across numerous gel samples and numerous experiments, and additionally does not depend on ramping hot-to-cold, cold-to-hot, or individually loaded samples for each temperature. We have recreated the figure below for ease. Briefly, the data we had plotted before had one aliquot of the sample per temperature. We have now included two new frequency sweeps of two different samples that were measured from low-to-high or high-to-low temperatures, allowing the materials to equilibrate at each temperature before measurements. The results indicate no temperature-induced changes in the plots regardless of going high-to-low or low-to-high. Slight differences in the magnitude of the moduli represent sample variation, but the overall temperature-dependent responses do not change. We have included all of this information in the methods section and in Supplementary Fig. 5.

Supplementary Fig. 5. Frequency sweep replicates with different temperature conditioning. **a**, Data plotted in the main text. Each curve was measured from one sample aliquot per temperature. **b**, In this case the sample was loaded at 12.5°C and measurements were taken from low to high temperatures, allowing the material to equilibrate at each temperature. **c**, This sample was loaded at 37°C and measurements were taken from high to low temperatures, allowing the material to equilibrate at each temperature.

Additionally, in the protocol, it is unclear what the order of temperature testing was for this gel? Hot to cold or cold to hot? A critical control experiment is needed to show that this does not matter, and that their results are reproducible. Regardless, even if they can show that the results are reproducible, the results are currently extremely underwhelming and insufficient to support their overarching hypothesis. For example, the experimental temperature range is very narrow, just 25°C. The authors explain that this is their experimental limit, but do not provide a reason for this limitation. Is it evaporation and/or dehydration? If so, they authors could try to seal the edge of the rheometer tool with mineral oil to prevent evaporation. This should easily allow data to be obtained from at least 5°C to 35°C, but conceivably much higher to maybe 60-70°C. Hopefully the experimental results would be more convincing over this larger temperature range whereby an observable gel relaxation could be reached.

We thank the reviewer for their comments and have clarified our exact protocol into our methods section. In the original data, all gels were formulated in large enough batches to split into 3 samples, one for each temperature, in order to minimize differences arising from batch variation or from shear memory. We have now included the results from 3 other gel batches and have tested hot-to-cold and cold-to-hot experiments, as outlined above. The results in Supplementary Fig. 5, provided above, illustrate no difference in temperature-dependent behavior between these experiments.

During our studies we attempted temperatures up to 70 °C with mineral oil surrounding the gel itself (which required us to forgo a serrated bottom plate and instead just use the Peltier plate bottom) and mineral oil at the seams of the solvent trap. In the first case, we unfortunately could not access large frequency ranges and reliable data due to leakage of mineral oil into the sample during flow experiments and large amplitude measurements. In the second case, we still observed large amounts of evaporation after a frequency sweep going from 0.1-100 rad/s at both 50 °C and 70 °C. Upon cleaning the geometries, it was clear that a solid-like ring form around the gel-air interface in these experiments, causing concern regarding the rheological results we obtained, so we decided to limit our study to 37.5 °C. We also believe that 25 °C is an adequate range to

observe useful temperature dependence for these soft hydrogels as evidenced by the orders of magnitude changes in modulus observed for similar, physically crosslinked hydrogels in the same temperature ranges^{1,2}. We have now included rationale of our temperature range into the methods section and in the main text. With regard to the reviewer's comments on the relaxation range, we have performed the suggested stress relaxation experiments and discuss the results in greater depth below.

1. Tan, C. S. Y. *et al.* Distinguishing relaxation dynamics in transiently crosslinked polymeric networks. *Polym Chem-Uk* **8**, 5336-5343, doi:10.1039/c7py00574a (2017).
2. van de Manakker, F., van der Pot, M., Vermonden, T., van Nostrum, C. F. & Hennink, W. E. Self-assembling hydrogels based on beta-cyclodextrin/cholesterol inclusion complexes. *Macromolecules* **41**, 1766-1773, doi:10.1021/ma702607r (2008).

Alternatively, why did the authors not try a series of step-strain tests at different temperatures, whereby the gel relaxation could more easily be observed and fitted to a single mode Maxwell model? As it stands, the presented rheological data does not look any different from other slow-relaxing polymer-particle hydrogels (entropically dominated or not).

We thank the reviewer for this suggestion and have performed stress relaxation experiments illustrating the temperature-dependent relaxation behavior of the gels. From results reported in our new Fig. 5e-f and Supplementary Fig. 7, we observe that a single-mode Maxwell model decently fits to the relaxation behavior from 10-420 s, with slight underestimations at the longer time points. Although from the frequency sweeps and a corresponding Cole-Cole plot, it is quite apparent these materials are not simply Maxwellian, we can nevertheless use these fits of the stress relaxation data to calculate a relaxation time for comparison purposes. We want to again emphasize here and in our updated Extended Discussion that the accurate fit of the rheological data to a single-mode Maxwell is not critical or necessary for gaining understanding of the temperature-dependencies that we are studying. Indeed, while the dominant contribution to viscoelasticity observed in many supramolecularly crosslinked polymer networks is captured by a single-mode Maxwell model, the rheological data of these systems are rarely (if ever) accurately fit in their entirety (for example, see the work by Craig *et al.*^{1,2}, Sherman *et al.*^{3,4}, and Kramer *et al.*⁵). Further discussion of the results of these experiments and the Maxwell fit is outlined below.

1. Yount, W.C. *et al.* Strong Means Slow: Dynamic Contributions to the Bulk Mechanical Properties of Supramolecular Networks. *Angew. Chem. Int. Ed.* **44**, 2746-2748, doi: 10.1002/anie.200500026 (2005).
2. Yount, W.C. *et al.* Small-Molecule Dynamics and Mechanisms Underlying the Macroscopic Mechanical Properties of Coordinatively Cross-Linked Polymer Networks. *J. Am. Chem. Soc.* **127**, 14488-14496, doi: 10.1021/ja054298a (2005).
3. Appel, E.A. *et al.* Supramolecular Cross-Linked Networks via Host-Guest Complexation with Cucurbit[8]uril. *J. Am. Chem. Soc.* **132**, 14251-14260, doi: 10.1021/ja106362w (2010).
4. Tan, C. S. Y. *et al.* Distinguishing relaxation dynamics in transiently crosslinked polymeric networks. *Polym Chem-Uk* **8**, 5336-5343, doi:10.1039/c7py00574a (2017).
5. Feldman, K. E. *et al.* Model transient networks from strongly hydrogen-bonded polymers. *Macromolecules*, **42**, 9072-9081, doi: 10.1021/ma901668w (2009).

Finally about Figure 4, it is unacceptable in its current form. As readers, we are supposed to be able to look at this data and easily track changes in the loss and storage moduli as a function of temperature; however, the resolution of the current plot is way too small, and the data symbols are way too big to discern any trends in these data. Again, my personal suggestion would be to try to measure and plot equivalent stress-relaxation data instead, since the proposed unique temperature-induced changes to gel relaxation time should be more easily confirmed.

We appreciate the reviewer pointing out our oversight and the uncertainty in the plots. We have replotted Figure 4 by reducing the axes values and reducing the symbol sizes to increase resolution. We have also included large-scale plots in the extended data sections to display the points with even greater clarity (Supplementary Fig. 4, reproduced below). We have also performed the stress relaxation experiments suggested by the reviewer and have included a zoomed-in plot in the extended data (Fig. 5e,f and Supplementary Fig. 7) and only display every 30th point to ensure the data is as clear as possible.

Supplementary Data Fig. 4. Larger plots of frequency and amplitude sweeps. Plots from Figure 4 are replotted here at higher resolution and size to more easily visualize differences between curves.

Along the same line of reasoning; the lack of a single mode Maxwell model fit to the experimental frequency sweep data is a fatal flaw to this study. The idea that the material behaves according to this model is fundamental to the support of their theory, and currently appears to be unfounded. If the authors can in fact collect frequency sweep data (or stress-relaxation data) and successfully fit it to a

single-mode Maxwell model, and actually show a clear increase in the network relaxation time as a function of increasing temperature, then the validation of the central hypothesis of the paper would be significantly enhanced.

[As a side-note; even if the authors can collect such data, I would be surprised if the data in fact fits a single-mode Maxwell model, given the multi-sticker nature of the PNP network crosslinks. Data from similar types of gels published in the literature at least suggests significant broadening of the resulting relaxation spectra. This shouldn't change the overall support of the entropy-dominated crosslink behavior proposed by the authors though.]

We chose a single-mode Maxwell model because it is the simplest starting model used to describe viscoelasticity in associative networks and commonly used to describe physical networks such as the CB gels we discuss in this manuscript. In many of these materials, the terminal relaxation region of the Maxwell model does not fit the experimental data well, but Maxwell is nevertheless still a reasonable starting point for modeling the data especially when there is a single G' plateau and G'' inflection point observed in the frequency range tested.

Ultimately, as the reviewer alludes to, an accurate fit of the single mode Maxwell model is not critical for our hypothesis on the entropy-dominated crosslinking behavior. However, we think the reviewer's comments on stress relaxation experiments are critical and we have performed the suggested experiments with the results now shown in Fig. 5e-f and enlarged in Supplementary Fig. 7. The single mode Maxwell fits demonstrate a negligible change in the τ relaxation time from 145 s at 12.5 °C to 108 s at 37.5 °C. In contrast, the enthalpy-driven CB[8]/MV/Np network decreases from 0.91 s at 10°C to 0.075 s at 30°, corresponding to a ~92% decrease in the relaxation time. The CB[8]/MV/Np relaxation times are taken as the crossover point of G' and G'' and calculated from the TTS shift factors reported in literature. The contrasting responses are particularly stark when the relaxation times for both systems are normalized to their respective relaxation times at 25 °C and plotted against temperature, as shown in Fig. 5f. This observation contrasts the prediction in Fig. 5a, which suggests there should be an increase in the relaxation time as the frequency sweep curve shifts leftwards towards lower frequencies.

The negligible decrease of the relaxation time apparent in the entropy driven PNP network indicates that additional relaxation mechanisms with different temperature dependencies may exist within the PNP networks that are not captured by the dilute ITC experiments. While slow relaxing contributions such as trapped strands and entanglements are enthalpy-driven and decrease relaxation times at higher temperatures, the entropy-driven PNP interactions work to counteract those temperature-induced changes. This hypothesis is corroborated by the negligible temperature dependence of stress relaxation in the PNP networks ($E_a \sim 2.5k_B T$ at 25°C) compared to the enthalpy-driven CB/MV/Np-based networks ($E_a \sim 36.5k_B T$ at 25°C). Moreover, our model predictions for an exothermic, entropy-driven network demonstrate this "dampened" reduction in temperature-induced relaxation times (Supplementary Fig. 1). The stress relaxation data put into the context of this model prediction suggests our dilute ITC experiments may not fully encapsulate the enthalpy-driven (and exothermic) components of trapped strands or entanglements that might exist in the concentrated case. We have expanded our discussion of the calculated predictions from our model and have included all of this discussion in the main text to accompany the new Fig. 5 e,f.

Fig. 5. Effect of temperature-induced changes in k_d on frequency-dependent rheometry. **a**, Calculated frequency-dependent G'/G_0 curves for PSNP/HPMC- C_{12} gels. **b**, Calculated G'/G_0 values taken at 10 rad/s show good agreement with experimental data and demonstrates negligible changes in modulus with elevated temperatures. **c**, Calculated frequency-dependent G'/G_0 curves for representative enthalpy-driven CB[8]/MV/Np hydrogels using literature values for crosslinking thermodynamics²⁹⁻³². **d**, Calculated G'/G_0 values taken at 10 rad/s show good agreement with literature data and demonstrates a significant decrease in modulus as temperature is elevated. **e**, Stress relaxation experiment with single mode Maxwell fits overlaid on the data points. Only every 30th point is displayed for clarity. **f**, Comparison of characteristic relaxation times between CB[8]/MV/Np and PSNP/HPMC- C_{12} normalized to the relaxation time at 25°C.

Supplementary Data Fig. 7. Stress relaxation data. **a**, A zoomed in plot of the stress relaxation data for each temperature overlaid with single mode Maxwell fits. Data are graphed by plotting every 30th point for clarity. **b**, A plot of the relaxation times determined from the single mode Maxwell fits.

Moving on to the other key experimental results in Figure 5, where the authors both compare their proposed entropic network with a known enthalpic network and relate these results to theory. A quick examination of their theoretical results show why this experiment has little value — the two networks have vastly different base relaxation times. In the entropic network, one observes little change in plateau moduli as a function of temperature, but that is predicted for both the entropic and enthalpic network if you remain in the plateau region far away from the relaxation time. In contrast, the enthalpic network is examined in the terminal relaxation region, so unsurprisingly there are large changes in the relative moduli as a function of temperature. In other words, if the dynamic shear measurements on the enthalpic network had instead been conducted at say 10^4 rad/s, then this materials viscoelastic moduli would also have been demonstrated to be temperature invariant. In essence, I do not see how this data looks any different from a comparison between “slow” and “fast” relaxing 'enthalpic' gel networks.

Instead, the theoretical predictions in Figure 5a actually clearly lay out what must be demonstrated experimentally to support the central hypothesis; an increase in relaxation time with increasing temperature. This behavior is predicted to be observed at low frequencies, which understandably can be hard to measure, but this frequency range needs to somehow be probed, either through a longer or higher temperature frequency sweep, or alternatively through another rheological measurement such as a stress relaxation, in order to clearly demonstrate the predicted unique entropy-dominated relationship between relaxation time and temperature.

We thank the reviewer for this insightful comment and suggestion. Figure 5 is first to illustrate that connecting crosslinking thermodynamics to G' is capable of predicting frequency sweep curves for enthalpy-driven networks such as the CB[8]/MV/Np system reported in the literature. The plots also demonstrate how an ideal entropy-driven network would change with temperature, assuming the dominant contribution to elasticity is due to the entropy-driven crosslinking interactions. We have included discussion and stress relaxation data to address the second half of this comment (please see our responses to previous comments).

Finally, a general note on the choice of vocabulary chosen throughout the text by the authors to describe the temperature-induced effects on the mechanical properties of the networks studied. Words such as ‘softening’ and ‘weakening’ needs to be treated with much more caution, as such words will

otherwise become confusing terms given the thermodynamic framework within which this study operates. Without proper explanation, these terms become ambiguous and leave the reader hanging as to what the author's actually mean, i.e. more liquid-like or networks with lower strength?

We thank the reviewer for pointing out these opportunities to be more precise in our language. We tried to reserve “soften” to represent decreases in G' and “weaken” for becoming more liquid-like and dynamic. We have included additional discussion early in the text to make this distinction explicit and we have also revised some sections to be explicit about liquid-like behavior when we mean that.

Reviewer #3 (Remarks to the Author):

I like this paper a lot. The idea that entropy can dominate in such situations is interesting but not without precedent. I will point you to the early work of Hooper and Schweizer (2005) where they showed that NP/polymer mixtures could phase separate under large enough attractive interaction due to bridging interactions. Why would bridging interactions give an entropic attraction? A recent paper by Sciortino et al., ACS Nano (2020) goes to the heart of this mechanism, on admittedly a different system. So, please read these previous works and see how your results fit in with this picture. Otherwise a nice contribution to the current literature.

We thank the reviewer for their kind words and useful literature suggestions. We have studied these suggested works and have included citations as well as discussion relating to these previous reports in relevant locations in the main text of our manuscript. From our rheometry results indicating an increase in caging energy and relatively no change in the modulus and relaxation times, it is highly suggestive that our PNP network is certainly not athermal, nor sterically stabilized. In the context of Hooper and Schweizer's work, our system can be thought of as bridging or the longer range telebridging. From our previous work examining the interparticle spacing and effects of polymer corona height, we hypothesize our system is most likely dominated by more local, bridging interactions¹. Interestingly, despite our D/d (D = particle diameter; d = monomer diameter) being estimated to be relatively large (>10), we do not see evidence of particle crystallization of any sort from preliminary, unpublished SAXS results.

Furthermore, while Hooper and Schweizer state that bridging is due to enthalpic stabilization of adsorbed polymer segments, these results are in polymer melts in absence of an additional solvent. In our system, water represents the third and most abundant component, which greatly affects entropic contributions towards desolvation of polymer chains as they adsorb to the nanoparticle surface. Interestingly, the recent paper by Sciortino et al. suggests an additional, combinatorial entropy contribution towards polymer-particle binding. Although the system used is indeed different from ours, based on our interparticle spacing and polymer corona height measurements, there is significant overlap of polymer coronas¹. We hypothesize that this overlap can offer the distinct intra- and inter-particle binding options to adsorbed chains, and therefore providing additional entropy gains. Unfortunately, our dilute-case ITC experiments cannot distinguish between these contributions, but it is certainly an interesting point to explore further, perhaps via additional simulation.

We have included these citations and brief discussion of these contributions into the main text.

1. Yu, A. C., Smith, A. A. A. & Appel, E. A. Structural considerations for physical hydrogels based on polymer–nanoparticle interactions. *Molecular Systems Design & Engineering* **5**, 401-407, doi:10.1039/C9ME00120D (2020).

Reviewers' comments:

Reviewer #1 (Remarks to the Author):

The authors have responded satisfactory to my criticism. A single point to make is that although I understand the reverse PMF experiment is not feasible due to technical constraints the authors should at least acknowledge that due to the limitations of sampling the configurational space the affinity by the pulling experiment will represent the strongest upper bound. Nevertheless, I find that the manuscript is interesting and deserves publication.

Reviewer #2 (Remarks to the Author):

I have reviewed the revised manuscript and while the paper is improved, the experimental data is in my opinion still not as convincing as needed to support the novel theory proposed.

Specifically, the addition of the stress relaxation experiment and discussion makes the paper a lot better; however, in my opinion, for it to be “worthy” of Nature Communications, the authors really needed to show that the materials relaxation time of the 'entropy-dominated networks' actually significantly increased with increasing temperature (as predicted by their theory), not just stayed about the same (or actually even slightly decreased). Hence, I personally still find the experimental evidence unconvincing to support their own theoretical predictions, and (less important) I still find their figures difficult to read or derive insightful meaning from in relation to their theoretical framework.

Reviewer #4 (Remarks to the Author):

Referee report on “Physical networks from entropy-driven non-covalent interactions” by A. C. Yu *et al.*

The authors present a combination of experiments and theory related to a physical network (made from polystyrene nanoparticles and HPMC-C₁₂). As raised by the previous reviewers, the experimental data are limited. It is hardly a surprise that the association-dissociation equilibrium of physical crosslink formation is contributed by both enthalpy and entropy, since the equilibrium constant is related to the exponential of the free energy of association. Therefore, the number of physical crosslinks arises from both enthalpy and entropy and is generally temperature dependent (weak or strong, depending on the specificity of the association/dissociation). Analogously, the dissociation rate constant k_d depends on temperature, as given in the equation in figure 3b (and equation 10 in SI),

$$k_d = k_a \exp\left(\frac{\Delta H^o}{RT} - \frac{\Delta S^o}{R}\right) = A \exp\left(\frac{E_a}{RT} + \frac{\Delta H^o}{RT} - \frac{\Delta S^o}{R}\right), \quad (1)$$

where equation 11 in SI is used. Therefore, if the phenomenon is entropically dominated, k_d must be insensitive to temperature. However, the authors show in figure 3d that k_d for PSNP/HPMC-C₁₂ depends significantly with temperature, that is by an order of magnitude over a range of 100° C. This is in full disagreement with the primary theme of the manuscript.

Surprisingly, the authors claim that “In this work, we develop a mathematical relationship between crosslink interaction thermodynamics and bulk viscoelasticity of the resulting physical networks”. This claim is unfair. I would encourage the authors to glance at the textbook by F. Tanaka on Polymer Physics, and in particular the Chapter 9 on “Rheology of thermoreversible gels”. From section 9.2.1 from this chapter, $G'(\omega, T)$ is written (in the notation of the present manuscript) as

$$G'(\omega, T) = \nu_e k_B T \frac{(\omega/k_d)^2}{(\omega/k_d)^2 + 1}, \quad (2)$$

where ν_e is the number of effective strands which is controlled by the association/dissociation equilibrium and depends on the temperature. The frequency dependence of G' depends only on k_d for the single mode relaxation model used in this manuscript, where k_d is given above. Appropriate time-temperature superposition, analogous to that in figure 9.3 of the above mentioned reference should capture the temperature dependence of $G'(\omega, T)$. The authors appear to be unaware of such classical results.

Overall, I conclude that this manuscript is not ready for publication.

Reviewer #2

I have reviewed the revised manuscript and while the paper is improved, the experimental data is in my opinion still not as convincing as needed to support the novel theory proposed.

Specifically, the addition of the stress relaxation experiment and discussion makes the paper a lot better; however, in my opinion, for it to be “worthy” of Nature Communications, the authors really needed to show that the materials relaxation time of the 'entropy-dominated networks' actually significantly increased with increasing temperature (as predicted by their theory), not just stayed about the same (or actually even slightly decreased). Hence, I personally still find the experimental evidence unconvincing to support their own theoretical predictions, and (less important) I still find their figures difficult to read or derive insightful meaning from in relation to their theoretical framework.

We thank the reviewer for their time considering our response and new experiments. In the reviewer's original comments, a primary concern raised was that “the lack of a single mode Maxwell model fit...is a fatal flaw to this study” and that “actually show[ing] a clear increase in the network relaxation time as a function of increasing temperature” was crucial for the central hypothesis of the paper. Among the experiments we added in our response was a set of stress relaxation experiments that not only fit well to a single mode Maxwell model, but also illustrated temperature-dependencies of the relaxation times that match the expectations outlined in Fig. 1 based on the temperature dependence of the PNP interactions shown in Fig. 3.

The reviewer's most recent comments suggest that we must “show that the materials [sic] relaxation time of the ‘entropy-dominated networks’ actually significantly increased with increasing temperature (as predicted by their theory), not just stayed about the same (or even slightly decreased).” We apologize for not clearly stating what our model implies for these experiments in our previous revision and we have now included calculated values for the normalized relaxation time in Fig 5f for direct comparison between theory and experiment. We have also clarified in the main text and supplementary what the expected temperature-dependent behavior is for the two experimental systems we evaluate, including our PNP system (mildly endothermic, entropically-driven) and a previously reported supramolecular host-guest hydrogel system (enthalpically-driven). Indeed, the k_d values for the enthalpically-driven CB[8]/MV/Np system are expected to increase roughly 5-fold over the temperature range we evaluated, which gives rise to the large decrease in the normalized relaxation time observed experimentally as shown in Fig. 5f (reproduced below). In contrast, the k_d values for our entropically-driven PNP system are expected to decrease only 2-fold over the same temperature range, corresponding to a negligible change in the normalized relaxation time similar to what we show in Fig. 5f below. We have expanded our manuscript text to discuss the expectations for these experiments and show that there is good agreement between experiment and theory. We have reproduced this text here for convenience:

“The model fits exhibit a negligible change in τ from 12.5°C to 37.5°C for the PSNP/HPMC-C₁₂ networks (Fig. 5e,f and Supplementary Fig. 7). In contrast, τ values estimated from the crossover frequency from frequency sweep plots reported in the literature for the enthalpy-driven CB[8]/MV/Np network³⁰ demonstrated a 92% decrease from 10°C to 30°C. The significant difference in temperature-dependent behavior between these two systems is further emphasized when τ is normalized to τ_0 at 25°C (Fig. 5f). Moreover, these experimental

data match expectations based on the temperature dependence of the k_d values for these two systems described above (Fig. 3d). Indeed, the k_d values for the enthalpy-driven CB[8]/MV/Np system are expected to increase 5-fold over the temperature range evaluated experimentally, commensurate with the 12-fold decrease in the normalized τ values observed experimentally (Fig. 5f). While the trends observed in the experimental data match theoretical expectations, the subtle deviation between the calculated and experimental normalized τ values for the CB[8]/MV/Np system likely arises from additional polymer-related relaxation events observed in this experimental system that are not captured in a single-mode Maxwell model³⁰. In contrast, the k_d values for the entropy-driven PSNP/HPMC-C₁₂ system are expected to decrease only 2-fold over the temperature range evaluated here, corresponding to the negligible change in the normalized τ values observed experimentally (Fig. 5f)."

Normalized relaxation time vs temperature data from Figure 5. **e**, Stress relaxation experiment with single mode Maxwell fits overlaid on the data points where only every 30th point is displayed for clarity. **f**, Comparison of characteristic relaxation times between enthalpy-driven CB[8]/MV/Np materials and entropy-driven PSNP/HPMC-C₁₂ materials normalized to the relaxation time at 25°C. Plot includes calculated relaxation times (dotted lines) for comparison to experimentally derived relaxation times. While all experimental hydrogel systems are expected to exhibit some additional relaxation behaviors beyond the primary relaxation mode captured in a single mode Maxwell model (*e.g.*, trapped strands or entanglements), the results of these stress relaxation measurements combined with the other supporting experimental data (Figure 4a-d in the revised manuscript) suggest good agreement between the theoretical predictions and the experimental results.

We would also like to emphasize that this work used three different rheological experiments to evaluate our PNP hydrogel model: frequency sweeps (Fig. 4a, Fig. 5a-d), strain sweeps (Fig. 4c-d), and stress relaxation experiments (Fig. 5e-f). In particular, strain sweeps showed increased caging energies observed for our entropically-driven system with increasing temperature, as shown in Fig. 4d, which we believe is a crucial piece of data regarding the temperature dependence of our PNP system that complements the observations related to temperature dependence of the relaxation time for the PNP system. We have amended the discussion to more explicitly highlight the agreement among these three experiments. Ultimately, all of these experiments indicate that networks formed with the mildly endothermic, entropically-driven crosslinking interactions in our study are behaving according to theoretical predictions.

We believe it is also important to recognize that all polymer systems are expected to exhibit additional relaxation mechanisms arising from trapped strands or entanglements that will introduce other temperature dependent changes to network mechanics. These confounding factors make it extremely difficult to experimentally measure entropic effects even in covalently crosslinked hydrogel networks, which should theoretically exhibit entropic elasticity. This will be true also of the subtle entropic effects observed in what we believe to be a unique example of an entropically-driven physical hydrogel described here. However, the fact that the relaxation times observed in our system are invariant to temperature and that the decaging energies increase with temperature are strong indicators that the properties in our PNP system are dominated by entropic effects, consistent with our theoretical model. We have included additional discussion regarding the opportunity to design entropy-driving crosslinking interactions to counteract potential impacts of polymer relaxation modes to fine-tune the mechanical behavior of these materials:

“Slow relaxing contributions such as trapped strands and entanglements arising in the concentration regime within the network are expected to exhibit decreased relaxation times at higher temperatures. While such behaviors will exacerbate the temperature-dependent relaxation of enthalpy-driven CB/MV/Np crosslinking interactions, the entropy-driven PSNP/HPMC-C₁₂ crosslinking interactions work to counteract these temperature-induced changes⁶⁵ (Supplementary Fig. 1). This hypothesis is corroborated by the negligible temperature dependence of stress relaxation in the PSNP/HPMC-C₁₂ networks ($E_a \sim 2.5k_B T$ at 25°C) compared to the enthalpy-driven CB/MV/Np-based networks ($E_a \sim 36.5k_B T$ at 25°C)³⁰. Ultimately, these observations highlight the value of designing endothermic, entropy-driven crosslinking interactions to generate physical network materials exhibiting mechanics that are invariant within working temperature ranges.”

With regard to the reviewer’s final comment, we have included enlarged half-page versions of the rheological data in Supplementary Fig. 4 of our revised manuscript, which we believe should be large enough to read at-scale. We would be happy to provide larger versions of these plots if this will help readers more easily derive insightful meaning from the figures.

Reviewer #4

The authors present a combination of experiments and theory related to a physical network (made from polystyrene nanoparticles and HPMC-C₁₂). As raised by the previous reviewers, the experimental data are limited. It is hardly a surprise that the association-dissociation equilibrium of physical crosslink formation is contributed by both enthalpy and entropy, since the equilibrium constant is related to the exponential of the free energy of association. Therefore, the number of physical crosslinks arises from both enthalpy and entropy and is generally temperature dependent (weak or strong, depending on the specificity of the association/dissociation). Analogously, the dissociation rate constant k_d depends on temperature, as given in the equation in figure 3b (and equation 10 in SI),

$$k_d = k_a \exp\left(\frac{\Delta H^\circ}{RT} - \frac{\Delta S^\circ}{R}\right) = A \exp\left(\frac{E_a}{RT} + \frac{\Delta H^\circ}{RT} - \frac{\Delta S^\circ}{R}\right)$$

where equation 11 in SI is used. Therefore, if the phenomenon is entropically dominated, k_d must be insensitive to temperature. However, the authors show in figure 3d that k_d for PSNP/HPMC-C₁₂ depends significantly with temperature, that is by an order of magnitude over a range of 100 °C. This is in full disagreement with the primary theme of the manuscript.

We thank Reviewer #4 for their comments and insight into the work. Though, we believe that the Reviewer's comments suggest that there was a misunderstanding of the theoretical framework and we apologize for any lack of clarity in our discussion. The Reviewer comments that *"if the phenomenon is entropically dominated, k_d must be insensitive to temperature,"* but this statement is inconsistent with what we describe in the main text and Fig. 1 and so we have adjusted our discussion to avoid this misunderstanding. Briefly, it is clear from the equation reproduced in the Reviewer's comments that the temperature dependence of k_d lies on the magnitude and sign of the enthalpy term. Indeed, even entropically-driven systems (*i.e.*, $\Delta S > 0$ and $|T\Delta S| \gg |\Delta H|$) can exhibit multiple types of temperature-dependent behaviors based on whether the system is exothermic or endothermic, as outlined in Fig. 1 and Supplementary Fig. S1. This equation suggests that for a system to exhibit a k_d value that is completely insensitive to temperature, the enthalpy of binding must be 0 kJ/mol. As our experimental system is an endothermic, entropy-driven system, we would expect to see the k_d value decrease with temperature, as shown in Figure 3d. We then show in both theory and experiment that for a system exhibiting a modulus value that is already in the plateau region, temperature-induced decreases in k_d do not alter the observable modulus values. For this reason, we believe that the comment *"this [observation] is in full disagreement with the primary theme of the manuscript"* is inaccurate. We have adjusted the text to avoid this confusion and included an excerpt from the results section here:

"Given that the hydrogel formulations are in the rubbery plateau under the conditions evaluated, with nearly all crosslinks bound ($\vartheta \sim 1.00$), no substantial change is observed in $G'/(v_0 k_B T)$. These trends are exemplified when the experimental data is overlaid onto the calculated curves for $G'/(v_0 k_B T)$ values taken at a representative frequency of 10 rad/s (Fig. 5b)."

Surprisingly, the authors claim that “In this work, we develop a mathematical relationship between crosslink interaction thermodynamics and bulk viscoelasticity of the resulting physical networks”. This claim is unfair. I would encourage the authors to glance at the textbook by F. Tanaka on Polymer Physics, and in particular the Chapter 9 on “Rheology of thermoreversible gels”. From section 9.2.1 from this chapter, $G_0(\omega, T)$ is written (in the notation of the present manuscript) as

$$G'(\omega, T) = \nu_e k_B T \frac{(\omega/k_d)^2}{(\omega/k_d)^2 + 1}$$

where ν_e is the number of effective strands which is controlled by the association/dissociation equilibrium and depends on the temperature. The frequency dependence of G_0 depends only on k_d for the single mode relaxation model used in this manuscript, where k_d is given above. Appropriate time-temperature superposition, analogous to that in figure 9.3 of the above mentioned reference should capture the temperature dependence of $G_0(\omega, T)$. The authors appear to be unaware of such classical results. Overall, I conclude that this manuscript is not ready for publication.

We believe that the reviewer has misunderstood the theoretical aspects of the presented work, evidenced by their suggestions that the classical results presented in the textbook by F. Tanaka are the same as the ones presented here. Indeed, the reviewer states that ν_e is related to the equilibrium constant and should therefore be a temperature dependent value; however, as we explain in the main text and show explicitly in Figure 3c, the crosslink density ν_e (which we represent as $\nu_0\theta$ in our study) negligibly changes with temperature given its relationship to K_{eq} . This observation suggests that temperature-induced changes in K_{eq} are not ultimately responsible for the clear temperature-induced changes to network mechanics often observed experimentally for enthalpically-driven physical hydrogels. Moreover, from the referenced textbook, it is apparent that the relaxation rate (β) is presented in an Arrhenius form, which is exactly what we replaced in this work with an expression that takes into account both the enthalpic and entropic contributions to the crosslink disengagement dynamics. We show for the first time that time-temperature superposition can be captured by the thermodynamics of the non-covalently crosslinked networks and describe what we believe to be a unique example of an entropically-driven physical hydrogel. We have included excerpts that we have amended in the manuscript to clarify these points:

“While enthalpy-driven crosslinks, including CD/Ad and CB[8]/MV/Np, and entropy-driven PSNP/HPMC-C₁₂ interactions exhibit opposite trends in the temperature dependence of $\theta(T)$, the magnitude of these changes are negligible ($\Delta < 1\%$) across the entire accessible temperature range for all of the physical networks evaluated^{28-30,33,37}. This observation suggests that temperature-dependent changes in network viscoelasticity observed in many systems are not caused by changes in effective crosslink density as offered by classical results⁵⁷. In contrast, calculations of the temperature dependence of k_d based on the thermodynamic parameters for these crosslinks indicate that k_d changes orders of magnitude over the same temperature range (Fig. 3d).”

From the discussion section:

“Two terms are useful to model the mechanical behavior of these systems: (i) the plateau modulus $G_0(T)$ and (ii) the frequency dependence $g(\omega, T)$. Temperature-induced changes to the mechanics of physical network systems observed experimentally are often attributed to changes in the plateau modulus of the materials arising from changes in the crosslink density ν_0 within the networks, which we represent as $\nu_0\theta(T)$ in our study. Yet, we show with several prominent experimental physical hydrogel systems that the proportion of bound crosslinks

$\vartheta(T)$, which is dependent on K_{eq} of the crosslinking interactions, does not change substantially over the entire accessible temperature range for hydrogels (i.e., 0-100 °C). Instead, the frequency dependence of the mechanics is implicated in significant temperature-induced changes in modulus values. We show that replacing the Arrhenius construction of the relaxation rate $\beta(T)$ with the enthalpic and entropic contributions to the dissociative rate constant, k_d , can capture temperature-induced changes in mechanical properties of distinct physically crosslinked networks.”

From the Supplementary Information:

“in the Arrhenius construction of $\beta(T)$, elevated temperatures will only lead to an increase in $\beta(T)$ for positive E_a values, resulting in a rightward horizontal shift in the frequency-dependent rheology (Supplementary Data Fig. 1). This construction, while physically intuitive, washes out more specific examination of the interacting moieties and precludes the possibility of other temperature-dependent behaviors.”

REVIEWERS' COMMENTS

Reviewer #1 (Remarks to the Author):

I read the reviewers' comments and the author's responses.

I believe that the model of a single Maxwell mode is a simplification of a rich dynamic behavior that is very difficult to probe without diving into molecular level details.

The authors did respond that potentially more modes are needed (additional relaxation mechanisms) but it is hard to introduce such models solely on experiments (thus I originally supported that a more extensive simulation work on this subject would be more than welcome). In short, my answer is that the lack of a decrease in relaxation time using a single Maxwell mode provides sufficient evidence that entropic factors are important and potentially prevailing in the system studied. Therefore, if that is the only remaining concern from the reviewers, the manuscript could still make an important contribution and I support publication.

As a final comment, I do agree though with the reviewer #2 on his comments about figures, if the authors can add large scale versions in supplemental as they suggest, that would be helpful.

Reviewer #2 (Remarks to the Author):

I commend the authors for improving the clarity of the manuscript, especially as it pertains to the underlying theory.

However, my original and primary concern remains; based on the data presented I am not convinced that the authors are indeed observing significant entropic effects in the PNP system. I remain concerned that the clear experimental differences observed between the 'enthalpic' networks and the 'entropic' networks at the temperature range studied, are primarily driven by the dramatically different network relaxation timescale between the two networks. In other words, while polymer-NP entropically driven interactions might indeed be at play in the PNP system, making this claim based on a comparison with another network system with such a different relaxation timescale while keeping the temperature range studied the same, is (in my humble opinion) shaky ground to stand on. Rather than comparing these two systems within the same temperature range, I believe it would be more appropriate to compare and analyze their behavior at the same effective "mechanical" regime, i.e. the same temporal "relative distance" away from the network-specific relaxation time well within the solid regime of both networks. I believe such a comparison could potentially provide more convincing experimental support for the proposed entropic origin of the PNP network thermal response.

Reviewer #3 (Remarks to the Author):

I have looked over the reviewer comments and have focused primarily on the authors' response to

referee 2. I dont understand referee 2 - yes a purely entropic system would behave exactly as they say. However, completely turning off enthalpy is basically impossible. So, in that sense, the authors are correct and the referee is being too critical. I think the paper should be published and then people can go ahead with more experiments and show if they are correct or if the story needs modification. At the moment the story is compelling and it is time to publish this paper

Reviewer #4 (Remarks to the Author):

I appreciate the efforts made by the authors to address my earlier remarks related to the temperature effects on 'entropically driven phenomenon'. I am afraid the response is still unsatisfactory in terms of the qualitative aspects of the claim and I regret to hold the same earlier conclusion.

Reviewer #1 (Remarks to the Author):

I read the reviewers' comments and the author's responses.

I believe that the model of a single Maxwell mode is a simplification of a rich dynamic behavior that is very difficult to probe without diving into molecular level details.

The authors did respond that potentially more modes are needed (additional relaxation mechanisms) but it is hard to introduce such models solely on experiments (thus I originally supported that a more extensive simulation work on this subject would be more than welcome). In short, my answer is that the lack of a decrease in relaxation time using a single Maxwell mode provides sufficient evidence that entropic factors are important and potentially prevailing in the system studied. Therefore, If that is the only remaining concern from the reviewers, the manuscript could still make an important contribution and I support publication.

As a final comment, I do agree though with the reviewer #2 on his comments about figures, if the authors can add large scale versions in supplemental as they suggest, that would be helpful.

We thank the reviewer for their consideration of our responses to the previous round of comments and their support for publication of this work. We agree that additional relaxation mechanisms will be necessary to comprehensively describe the complex behavior of these materials and have included additional text in the manuscript to discuss this more clearly. Regarding the figures, we have split Fig. 5 into two figures and increased the size of the plots throughout the manuscript to better display the data. We have attached the new Figures 6 below to highlight that we have sought to improve readability of the data by truncating some of the axis and expanding the plot size:

Fig. 6. Temperature-dependent stress relaxation experiments. a, Stress relaxation experiments with single mode Maxwell fits overlaid on the data points where only every 30th point is displayed for clarity. **b,** Comparison of characteristic relaxation times between CB[8]/MV/Np and PSNP/HPMC-C₁₂ networks normalized to the relaxation time at 25°C. Dotted lines represent calculated values from the model.

Reviewer #2 (Remarks to the Author):

I commend the authors for improving the clarity of the manuscript, especially as it pertains to the underlying theory.

However, my original and primary concern remains; based on the data presented I am not convinced that the authors are indeed observing significant entropic effects in the PNP system. I remain concerned that the clear experimental differences observed between the 'enthalpic' networks and the 'entropic' networks at the temperature range studied, are primarily driven by the dramatically different network relaxation timescale between the two networks. In other words, while polymer-NP entropically driven interactions might indeed be at play in the PNP system, making this claim based on a comparison with another network system with such a different relaxation timescale while keeping the temperature range studied the same, is (in my humble opinion) shaky ground to stand on. Rather than comparing these two systems within the same temperature range, I believe it would be more appropriate to compare and analyze their behavior at the same effective "mechanical" regime, i.e. the same temporal "relative distance" away from the network-specific relaxation time well within the solid regime of both networks. I believe such a comparison could potentially provide more convincing experimental support for the proposed entropic origin of the PNP network thermal response.

We thank the reviewer for their insight and believe their comments have helped improve the manuscript. The enthalpy-driven system we used as a comparison to our entropy-driven system was chosen due to the availability of frequency sweeps at varying temperatures (C.S.Y. Tan *et al. Polym Chem-Uk*, **2017**, *8*, 5336-5343). This same report describes two other formulations of the enthalpically-driven CB[8]/MV/Np system using higher molecular weight polymers that can be viewed as existing more within the same "mechanical regime" as our PNP system. While the crossover frequencies are very low and out of the typical observable frequency range, by examining the shear storage modulus values at 10 rad/s (the same as in our manuscript), it is apparent that even in these formulations the storage modulus drops ~1 order of magnitude across a similar temperature range (0-40 °C). This mechanical behavior is consistent with what we would predict for this enthalpy-driven system and clearly contrasts the near invariant mechanical properties observed for the PNP system.

For our analysis, we chose the CB[8]/MV/Np formulation comprising 90 kDa polymers to compare with our PNP system because the rheological behavior of this system was dominated by the supramolecular crosslinking interactions. In contrast, the formulations comprising higher molecular weight polymers (1300 kDa and 250 kDa) exhibited significant contributions of the polymers to the relaxation behavior in addition to the non-covalent interactions. In our studies, the rheometry experiments on our PNP system suggest that the HPMC-C₁₂ polymers do not significantly interact on their own and the mechanical properties of the PNP networks are dominated by the PNP crosslinking interactions. For these reasons, we believe the CB[8]/MV/Np formulation comprising 90 kDa polymers provided the most appropriate mechanical comparison.

Ultimately, the ideal study would compare two systems that are as similar as possible with the sole exception being the crosslinking thermodynamics. Nevertheless, we believe the simulations, calorimetry, and multiple rheological experiments with two hydrogels comprising very similar polymers provides ample support for our hypothesis.

Reviewer #3 (Remarks to the Author):

I have looked over the reviewer comments and have focused primarily on the authors' response to referee 2. I don't understand referee 2 - yes a purely entropic system would behave exactly as they say. However, completely turning off enthalpy is basically impossible. So, in that sense, the authors are correct and the referee is being too critical. I think the paper should be published and then people can go ahead with more experiments and show if they are correct or if the story needs modification. At the moment the story is compelling and it is time to publish this paper

We thank the reviewer for their analysis and supportive remarks regarding our description of additional enthalpic relaxation modes that are certain to exist in any system to some extent. We believe the mathematical relationship between crosslink interaction thermodynamics and bulk viscoelasticity that we describe, in combination with the experiments and simulations, provides an important contribution for understanding entropy-driven crosslinked networks. Future experiments will certainly help to fully expand our understanding of these concepts.